# Increased perception of the experience dimension of the animal mind reduces instrumental violence against animals

**Agnieszka Potocka**[1,2]*, **Maksymilian Bielecki**[1]

**1** Institute of Psychology, SWPS University, Warsaw, Poland, **2** Institute of Philosophy and Sociology, The Maria Grzegorzewska University, Warsaw, Poland

* apotocka3@st.swps.edu.pl

**Data Availability Statement:** All data are fully available on OSF: study preregistration: https://doi.org/10.17605/OSF.IO/6JKST, database, and syntax: https://doi.org/10.17605/OSF.IO/XMYV8, licensed under CC-BY Attribution 4.0 International.

## Abstract

In this study, we investigated whether the perception of animal experience capacities, enabling individuals to recognize animals as moral patients, decreases instrumental violence against animals. Additionally, we aimed to distinguish this effect from the influence of perceptions of agency capacities, referred to as anthropomorphization. To achieve this, we conducted an online experimental study ($N$ = 471, 54% women). Participants performed a manipulation task that increased their perception of the experience dimension of the animal mind and completed online questionnaires as part of a manipulation check to measure acceptance and intentions of instrumental violence against animals. Regression and mediation analyses revealed that increasing perception of the experience dimension of animal mind decreases instrumental violence against animals, particularly intentions to commit such violence, and this effect is unique and distinct from the effect of perception of the agency dimension, i.e., anthropomorphization. The key capacities in lowering violence were homeostatic emotions (pain, fear, hunger, and thirst) which indicate suffering that humans would want animals to avoid. However, when people perceive homeostatic emotions, increased perception of more complex capacities (anger, joy, pleasure, personality) and anthropomorphization do not result in an additional reduction in violence. We interpret these results to mean that people limit violence by using perception of animal experience capacities as pre-violation justification. These findings expand our knowledge about the functions of perception of experience capacities and demonstrate that people diminish animal experience capacities not only to rationalize violence but also as pre-violation justification to facilitate instrumental violence against animals.

## Introduction

Violence towards animals has remained a persistent and unsettling issue, presenting ethical and moral challenges. Despite the importance of this topic, the field is far from reaching a comprehensive understanding of the psychological mechanisms that underpin these relationships. In the current literature, authors [1–5] have discussed how people perceive or

**Funding:** This work was supported by the Animal Advocacy Research Fund, funded by Animal Charity Evaluators. Agnieszka Potocka (Principal Investigator) was the author who received the award. Agnieszka Golec de Zavala and Maksymilian Bielecki were declared as key collaborators. The grant does not have a specific grant number and was signed on September 10, 2020. Funder name: Animal Charity Evaluators https://researchfund. animalcharityevaluators.org/. Since funder may have a vested interest in the research reported in the enclosed paper to eliminate any role of this organization in the research, analysis, or paper preparation process, the study was preregistered before collecting data, and funding was not contingent upon results. All funds or sources of support received during this study have been described in accordance with the requirements in the online author's guide at http://journals.plos.org/ plosone/s/submit-now. The funders had no role in study design, data collection and analysis, decision to publish, or preparation of the manuscript. There was no additional external funding received for this study.

**Competing interests:** I have read the journal's policy and the authors of this manuscript have the following competing interests: To conduct this study Agnieszka Potocka received Animal Advocacy Research Fund, funded by Animal Charity Evaluators. This does not alter our adherence to PLOS ONE policies on sharing data and materials. The data and protocol are licensed under CC-By Attribution 4.0 International.

comprehend animals' mental capabilities as an important contributing variable. Some academics have indicated that beliefs or perceptions of mental abilities serve to rationalize, justify, or constrain instrumental violence against animals [6–11]. However, some inconsistencies persist. Potocka and Bielecki [12] showed that perception of experience capacities (e.g., fear, pain, hunger, emotions, personality) is negatively associated with the acceptance of violence against animals, while anthropomorphization, when simultaneously accounted for, exhibits no association or a positive one. Conversely, other studies found that anthropomorphization reduced instrumental violence related to meat consumption [10, 11, 13, 14], but these studies did not account for experience capacities.

Therefore, there is still a gap in our knowledge. It is unclear whether (1) the perception of experience capacities alone reduces violence against animals while controlling for anthropomorphization, (2) they both simultaneously reduce violence, or (3) the perception of experience capacities is only correlated with instrumental violence against animals.

This study fills this gap and expands our understanding of whether perceptions of animals' experience capacities to feel basic psychological states (i.e., fear, hunger), emotions, and possess consciousness and personality (the experience dimension of animal mind) [15–17] influence instrumental violence against animals and whether this is unique and distinct from the effect of anthropomorphization.

Following Waytz, Gray et al. [17], we anticipate that an increase in the perception of the experience dimension of an animal mind would curtail instrumental violence against animals since it determines the animals' high moral patient status and their right to remain unharmed. At the same time, we propose that this observed effect is separate from the attributed agency aspect of an animal mind, often described as moral agency, anthropomorphization, or personification [15, 18–22].

## Instrumental violence against animals

People have been using animals for their purposes since the domestication of animals [23–25]. The term "use" often means instrumental violence, that is, breeding animals to produce meat, leather, dairy products, or eggs [4]. During such food production, animals are enslaved from birth, often spend their lives in cages, have poor welfare, and are eventually killed as juveniles. Several studies explored this topic, investigating attitudes towards animal use [26–28], treatment [29, 30] or welfare [31, 32]. The authors highlight that political and social structures rooted in religion, culture, and traditional worldviews, including conservatism and anthropocentrism [33–35] facilitate the instrumental use of animals. Individual factors related to our relationship with animals could be classified into two broad categories: relatively stable dispositions or factors more susceptible to change. Relatively stable dispositions include factors negatively related to violence against animals, such as empathy [27, 36–39], moral idealism [40, 41], the moral foundation of care [4, 12], values such as universalism [42], as well as positively related factors: dark triad traits [43–45], moral relativism [40, 41], the moral foundation of authority [4, 12], social dominance orientation [46], and values such as tradition, obedience, and security [42, 47, 48].

The second set of individual factors linked with violence against animals contains those that are more susceptible to change over time due to varying situational or social influences. This includes aspects like individual or collective goals or the perception of animal attributes. Goals serve as the catalyst for social interaction, and when these goals are instrumental and focused on acquiring benefits, they can trigger violence [49–51]. Individuals engaged in the primary industry, as well as animal breeders, farmers, and dog trainers whose goals are financial, are more prone to committing or condoning violence compared to the general

community samples [49–53]. Conversely, those who connect over animal rights or welfare issues typically oppose violence [54, 55]. Another adaptable factor that plays a significant role in our relationship with animals is our perception of certain animal traits, such as their infant-like appeal, cuteness [56–58], or our perception of animal consciousness, as elaborated below.

## Perception of animal mind

According to the literature, early research focused on the assumption that people's beliefs about animal minds are constant and on analyzing these with regard to animal consciousness, feelings, emotions, thinking, and ability to solve problems [2, 3, 28, 59]. Further research, mainly in the field of animal welfare, focused on the concept of sentience [1, 60–62]. Both views showed that people rated mammals the highest, followed by birds, fish, and insects. However, people favor companion animals, e.g., dogs and cats, above other similar species [2, 63, 64].

Another perspective shows the instability of our beliefs [16, 17, 65]. People fit their perception of the animal mind to the situation, intention, and socially relevant goals. Meat eaters (instrumental goal—food) have a lower perception of the mental capacities of animals [4, 7, 66]. Similarly, animal mind attribution was lower when animals were being taken to an abattoir to be killed rather than to a pasture [6]. Such mechanisms were interpreted as rationalization by dementalization or infrahumanization. Animals were described as entities that have less capacities to feel secondary emotions or generally have inferior minds when humans harm or use them [6–8, 67]. This suggests that people adjust their perception of the animal mind to fit the situation or current instrumental goals [67].

A slightly different perspective is proposed in research on anthropomorphization or personification. Both refer to capacities stereotypically and traditionally understood as unique to humans [20, 22]. Anthropomorphization is described as "attributing a humanlike mind to nonhumans" [p. 221, 22], in particular, intentionality, emotions, thoughts, feelings, motivations and beliefs, identity, understanding phenomena, and the ability to predict the behavior of others [20, 22, 68]. Personification is defined in a similar fashion—mainly in law, philosophy, or in literature [21, 69–71]. Rooted in this perspective, previous studies presented anthropomorphization as a factor that limits violence or protects animals against violence [10, 11]. No matter how important anthropomorphization's role in limiting violence is, it is certainly not the only aspect of the animal mind perception regulating our moral choices [16, 17, 65].

## Role of mind perception in morality

Within the context of moral typecasting, as delineated by Gray & Wegner [16] individuals perceive two roles in moral situations: the moral agent who helps or harms, and the moral patient who is helped or harmed [16]. Gray and Wegner [16, 65] suggest that when the object is perceived as a moral agent (perpetrator of good and evil), he/she/it is also perceived as possessing attributes linked to reasoned actions such as self-control, planning, thinking, and communication. Conversely, when the entity is viewed as a moral patient, the recipient of good or evil, it is perceived as possessing capacities to feel basic psychological states such as fear, hunger, thirst, primary and secondary emotions like pride, joy, sadness, happiness, and also having consciousness and personality [15, 16, 65, 72].

However, in our study, we leverage the theoretical framework proposed by Waytz, Gray et al. [17], which explores the causes and effects of mind perception in relation to morality. This approach represents the perspective of a potential perpetrator (moral agent, i.e., an adult person with motivation for social connection and control) rather than an observer, as characteristic of Gray and Wegner's theory of dyadic morality [65]. We chose this framework since previous research indicates that when individuals contemplate using others without

appropriate compensation or inflicting harm or death for personal benefit, they acknowledge the likelihood of being negatively perceived by the community and deemed as wrongdoers [73–76], expecting feelings of guilt or embarrassment [77–79]. This arises because an observer who witnesses a victim in distress (a moral patient) seeks to complete the moral dyad by identifying a perpetrator to blame [65, 80]. Meanwhile, people (perpetrators) desire to maintain their moral self-image, be perceived positively by others, and prefer to avoid negative social repercussions and punishment [17, 73–76, 81]. As such, when people (perpetrators) exploit animals, they view themselves as potential moral agents, and if they intend to inflict violence, they seek justifications to enable their actions, similar to human-human relationships [82–85]. Therefore, following Waytz, Gray et al. [17], people attempt to diminish the mental capacities of their potential victim by making a shift from moral act (the victim is a moral patient) to a non-moral act (the victim is not a moral patient but an inanimate product, object, or thing). This allows for the dismissal of moral or legal responsibility for causing harm [17, 86]. By contrast, when people aim to avert violence, they utilize high moral patient status (high experience capacities) as a justification, as it attributes rights to the victim [17, 86].

In summary, we argue that regulation of perception of animals' experience capacities precedes decision-making and limits animal harm [17] since, from the perspective of moral agents (people as potential perpetrators), objects with higher moral patient status have more moral rights. We assert that instrumental violence against animals is prevalent among adults. We exploit animals in our everyday lives, and adults are aware of various justifications and rationalizations for causing harm to animals, particularly the belief that animals are objects devoid of emotions or experiences—ideas that are deeply embedded in our culture, religion [35, 87, 88], and an anthropocentric worldview [67, 89–93]. As a result, when individuals repeat these behaviors and have previously experienced social reactions to violence, they easily resort to excuses or pre-established rationalizations as pre-violation justifications or reasons to limit violent behavior [86, 94]. In such instances, the perception of mind, which determines the status of a moral patient, is used as a pre-violation justification (low) or a reason to shield animals from violence (high).

However, there are two contending explanations for pre-violation justification concerning two-dimensional mind perception [15]: granting the status of moral patient based on perception of the experience dimension and anthropomorphization [10, 11, 20] based on perception of the agency dimension.

## Current research

To address the research question, we implemented an experimental study. We argued that enhanced perception of an animal's capacity for experience reduces instrumental violence towards animals. This is based on the assertion by Waytz, Gray et al. [17] that an entity with a higher capacity for experience enjoys a higher moral patient status and thus merits protection. Nevertheless, we accounted for the agency aspect of animal consciousness for two reasons. Firstly, perceptions of an animal's capacity for experience and agency are intimately linked [12] and are hence presumed to have similar roles. Secondly, the same group of capacities characterizes both the agency dimension and anthropomorphization, which could potentially be animal-friendly [10]. As such, it is imperative to discern whether the perception of the experience dimension (as a moral patient) uniquely reduces instrumental violence against animals compared to the perception of the agency dimension (as anthropomorphization).

We controlled for all moral foundations [75, 95] since in previous studies, authors [4, 12] revealed the link between those foundations and both the perception of the animal mind and instrumental violence.

We tested one (preregistered) hypothesis: that increasing perception of the experience dimension of the animal mind (perception of experience) reduces instrumental violence against animals, and this effect is distinct from the effect of perception of the agency dimension (anthropomorphization).

## Methods

We conducted an experimental study with one factor on three levels. The manipulation was tested in a pilot study (see S1 Appendix) and it was expected that manipulation would increase perceptions of experience capacities in the Experience 1 and Experience 2 groups, resulting in lower instrumental violence against animals in comparison with the Control group. The study was preregistered on OSF: https://doi.org/10.17605/OSF.IO/6JKST (database: https://doi.org/10.17605/OSF.IO/XMYV8). Sample size estimation was performed using MedPower [96]. For path a X→M β = 0.42 (pilot results), path b M→Y β = −0.24 [12], direct path c' X→Y β = −0.03. For the test's power of 0.80 and p < 0.05, the minimum sample size was 455 people for a total effect of β = −0.131. We checked experience and agency capacities as well as moral foundations.

The study was approved by the Ethical Review Board at the SWPS University of Social Sciences and Humanities, Faculty of Psychology in Warsaw (no. 68/2021, 2/2022; the numbers of approval decisions are provided for both the pilot study and the main study since they were part of the same application).

### Participants and procedure

In February 2022, 664 Polish participants were recruited from the research panel of SW Research. Only adults over 18 years old were allowed to participate in the study. We required a sample of 52% female, 48% male, 60% urban, 40% rural, and 50% participants avoiding any animal products. In the online survey, a description of the study and masked information about the purpose of the study (this study explores the importance of emotions in agricultural management) were displayed. Afterwards, participants provided their informed consent by selecting the appropriate response ("Yes" or "No") to the question "Do you agree to participate in the study?". If a participant refused to participate (choose "No"), they could not continue with the study and were excluded from it (the answer is documented electronically, and the Ethical Review Board at the SWPS University of Social Sciences and Humanities, Faculty of Psychology in Warsaw, approved this consent procedure). Next, each participant was randomly assigned to one of three groups: control, Experience 1, Experience 2, and performed a math task and a manipulation task, completed AMQ-PL, and after that, performed an Animal Matrix task and completed MFQ-PL. Finally, the real goal of the study was revealed. The authors had no access to information that could identify individual participants during or after data collection.

Before the analyses, the quality of the collected data was verified. We excluded those respondents who answered incorrectly on control items and provided identical answers to all items on at least one of the questionnaires used (*n* = 55), whose answers represented other forms of violence than instrumental violence or suggested that the respondent either did not understand the task or gave a random answer in the Animal Matrix task (items 12–14, *n* = 12). Since some participants took a long time to complete the survey, we excluded those who ran the survey for 30 min or more. The final sample of 471 participants was aged between 18 and 83 years old (M = 45.55, SD = 16.91), 53.9% were women, 60.9% lived in a city, 39.1% lived in the country, 46.5% had higher education, 27.3% had secondary and post-secondary education, and other responses were rare (4.7% vocational, 1.5% primary education). A large majority (87.3%)

had pets during the last five years, and 22.1% declared that their work is associated with animals.

## Experimental manipulation

We used manipulation tested in the pilot study (priming experience capacities of animal mind, see Appendix). All participants read texts that were assigned to their specific experimental condition. In the control group, the text was about colors. In Experience 1, it was about cows' and pigs' capacities to feel pain, fear, hunger. The text assigned to Experience 2 was about cows' and pigs' capacities to feel anger, joy, and pleasure, and about the animals' personalities. Next, the participants were asked to write a short post for Facebook or Twitter about the text they read and to complete the Animal Mind Questionnaire (AMQ-PL, [12]) based on the Mind Survey [15]. We used the AMQ-PL subscale of experience ($\alpha = 0.90$) to verify the effectiveness of the manipulation and the agency subscale ($\alpha = 0.89$) to control for anthropomorphization. The experience subscale contains items on hunger, fear, pain, pleasure, rage, desire, personality, consciousness, pride, embarrassment, and joy. The agency subscale uses items on morality, memory, emotion recognition, planning, self-control, communication, and thought. Participants instructions were the following: "Using a scale of 0 (not at all) to 6 (very much), please tell us how capable you think horses, cows, goats, sheep are of:" and provided with descriptions of 18 mental capacities from the original scale, e.g., "understanding how others are feeling", "feeling afraid or fearful", "telling right from wrong and trying to do the right thing".

## Measures

**Instrumental violence.**  We measured the intentions (violence intentions) and acceptance (violence acceptance) of instrumental violence against animals. To measure intentions, we prepared an Animal Matrix task based on tools elaborated by Tajfel [97] (matrix 4), Sachdev & Bourhis [98], and Sidanius et al. [99]. The participants performed the task of planning an animal husbandry budget.

The participants were asked to divide an expected income of PLN 23,000 from farm animal sales between animal-related costs and their own profits. Animal costs refer to maintenance and providing food, facilities, veterinary care, and satisfying other needs that increase animal welfare for 10 animals (pigs or cows). Participants were informed that the minimum cost of breeding one animal is PLN 1,150 (11,500 for 10 animals) and that their profit from breeding will be the total income from sales less costs. The task included 14 suggestions on how to distribute the expected income (PLN 23,000). Three items did not represent instrumental violence and served as control items. These were the two lowest-cost (PLN 5,000 and 6,000, items 13 and 14) suggestions that resulted in the death of the animals, and one cost suggestion (PLN 7,000, item 12) that resulted in the same income as a higher animal cost (PLN 8,000). The remaining 11 suggestions differed in the cost-profit ratio and represented instrumental violence. When participants voted for more money for the animals, their profit was lower. However, even if the participants chose the maximum amount of money for the animals (PLN 18,000), they would still earn a good profit (PLN 5,000). The two middle suggestions represented a similar distribution, with animal costs being equal to the profit earned by the participant as a breeder. Less animal costs meant higher instrumental violence against animals.

To measure violence acceptance, we used nine items from a 23-item Polish version of the Animal Attitude Scale [12, 29, 100] (AAS-PL). Participants referred to items about acceptance of animal use, e.g., "I think it is perfectly acceptable for cattle and hogs to be raised for human consumption" and "Breeding animals for their skins is a legitimate use of animals". To create a violence acceptance index, we calculated the mean of nine items referring to violence against

farm animals (6, 8, 9, 13, 14, 21R, 22, 23, α = 0.88). Participants' responses were recoded, and higher results reflected higher Violence Acceptance (in contrast to the method in the original scale). The AAS-PL was rated on a five-point scale.

**Controlled variable.** We controlled for all moral foundations. They were measured according to the Polish adaptation of the Moral Foundations Questionnaire (MFQ-PL) [101, 102] (care α = 0.72, authority α = 0.70, loyalty α = 0.68, purity α = 0.73, fairness 5-item scale α = 0.75, in order to increase fairness subscale reliability, item 27 was excluded, for a 6-item subscale α = 0.66). MFQ-PL includes six items per foundation and is measured on a six-point scale.

Masking task. Participants solved four simple math tasks.

## Results

First, we calculated the means of variables (Table 1). Next, to confirm the effect of manipulation, we calculated contrast tests in OLS model for perception of experience, anthropomorphization and all moral foundations. We compared the experimental groups with the control group (Contrast 1) and the experimental groups with each other (Contrast 2) (Table 1).

Results confirmed higher perceptions of experience in the experimental groups (Experience 1 + Experience 2) versus control; however, this is accompanied by higher anthropomorphization (Contrast 1). A contrast test between the experimental groups (Contrast 2) revealed higher perception of experience in Experience 2 than in Experience 1, without differences in anthropomorphization. In both contrasts, there were no differences in any of the moral foundations.

Next, to test the main hypothesis, we used a mediation approach [103, 104] and estimated two parallel mediation models (Figs 1 and 2), which were estimated using the PROCESS macro (v4.2, Model 4) [105] for SPSS. The variable group was an independent multicategorical variable (X), perception of experience (moral patient) ($M_1$) and perception of agency (anthropomorphization) ($M_2$) were mediators, violence intentions were the dependent variable (Y) in model 1, and violence acceptance (Y) in model 2. We used Huber-White heteroscedasticity-consistent inference and the coding system (Helmert) that allowed us to compare the experimental groups (Experience 1 + Experience 2) with the control group ($X_1$) and the experimental groups with each other (Experience 2 vs. Experience 1, $X_2$).

**Table 1. Means and differences in the status of moral patient, anthropomorphization and moral foundations between groups.** Results of contrast tests.

| Measure | Total | | Control N = 149 | | Experience 1 N = 162 | | Experience 2 N = 160 | | Contrast 1 | | | Contrast 2 | | |
|---|---|---|---|---|---|---|---|---|---|---|---|---|---|---|
| | *M* | *SD* | *M* | *SD* | *M* | *SD* | *M* | *SD* | *Difference* | *SE* | *p* | *Difference* | *SE* | *p* |
| Perception of experience (moral patient) | 3.92 | 1.08 | 3.71 | 1.21 | 3.87 | 1.06 | 4.18 | 0.91 | **0.32** | **0.11** | **0.003** | **0.31** | **0.12** | **0.010** |
| Perception of agency (anthropomorphization) | 2.62 | 1.30 | 2.41 | 1.30 | 2.61 | 1.34 | 2.82 | 1.23 | **0.30** | **0.13** | **0.019** | 0.21 | 0.14 | 0.145 |
| Care | 5.18 | 0.62 | 5.12 | 0.68 | 5.18 | 0.57 | 5.23 | 0.59 | 0.09 | 0.06 | 0.151 | 0.05 | 0.07 | 0.484 |
| Fairness | 5.11 | 4.62 | 5.08 | 0.67 | 5.13 | 0.63 | 5.12 | 0.61 | 0.07 | 0.06 | 0.272 | −0.01 | 0.07 | 0.838 |
| Authority | 3.80 | 0.83 | 3.79 | 0.84 | 3.82 | 0.81 | 3.80 | 0.84 | 0.02 | 0.08 | 0.790 | 0.02 | 0.09 | 0.855 |
| Loyalty | 4.09 | 0.73 | 4.08 | 0.74 | 4.06 | 0.70 | 4.12 | 0.76 | 0.01 | 0.07 | 0.846 | −0.05 | 0.08 | 0.519 |
| Purity | 4.27 | 0.85 | 4.21 | 0.87 | 4.31 | 0.88 | 4.29 | 0.80 | 0.09 | 0.08 | 0.308 | −0.02 | 0.09 | 0.789 |

Note. *N* = 471. Bold font indicates statistically significant results.

Contrast 1 = both experimental groups together (Experience 1 and Experience 2) vs. the control group; contrast 2 = Experience 2 vs. Experience 1.

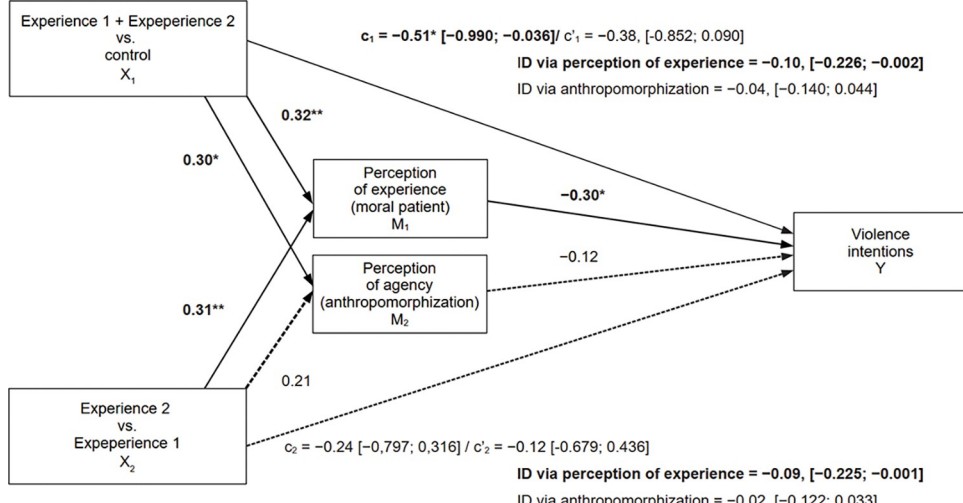

**Fig 1. Model 1.** Regression model depicting the effects of group and the mediating role of perception of experience and anthropomorphization on violence intention. Numerical values represent non-standardized regression coefficients. $N = 471$. $R^2 = 0.04$, $F(4,466) = 4.41$, $p = 0.002$. Bold font indicates statistically significant results. $^*$ $p < 0.05$, $^{**}$ $p < 0.01$ $^{***}$ $p < 0.001$.

## Discussion

The results (Figs 1 and 2) supported our hypothesis that perception of experience (moral patient) reduces instrumental violence against animals, and this effect is distinct from the effect of perception of agency (anthropomorphization). Our hypothesis was supported by revealing the total and indirect effects for the outcome variable violence intentions (Fig 2) when comparing both experimental groups with the control group ($X_1$). Moreover, referring to the recommendations of Shrout & Bolger [104] concerning interpretation of mediation effects in studies with an experimental design, indirect effects of comparison between both

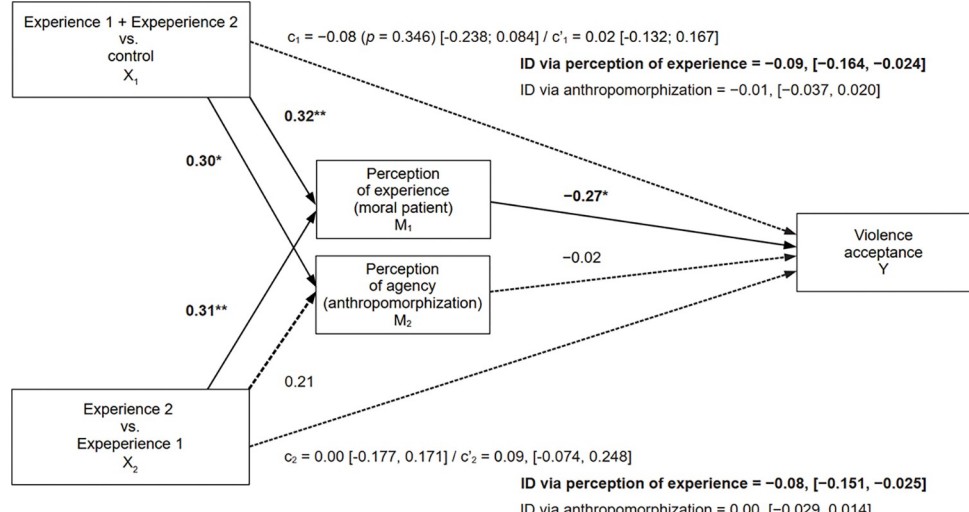

**Fig 2. Model 2.** Regression model depicting the effects of group and the mediating role of perception of experience and anthropomorphization in violence acceptance. Numerical values represent non-standardized regression coefficients. $N = 471$. $R^2 = 0.15$, $F(4,466) = 19.53$. Bold font indicates statistically significant results. $^*$ $p < 0.05$, $^{**}$ $p < 0.01$ $^{***}$ $p < 0.001$.

experimental groups and the control group ($X_1 \rightarrow M_1 \rightarrow Y$, Fig 2), and comparison between experimental groups ($X_2 \rightarrow M_2 \rightarrow Y$, Figs 1 and 2) also suggest support for our hypothesis even if the confidence interval of the total effects ($X_1 \rightarrow Y$ and $X_2 \rightarrow Y$, Figs 1 and 2) included zero.

Our interpretation of the influence of perception of experience on instrumental violence aligns with the theoretical framework presented by Waytz, Gray et al. Waytz, Gray et al. [17]. As individuals view themselves as potential moral agents (those willing to exploit animals), the perception of animal experience capacities enables them to assign animals the status of moral patients, along with related moral rights. Therefore, when perception of experience is low, animals have fewer moral rights, making people more likely to harm them [17]. These findings are congruent with Shalvi et al. [86] and Lowell [94], suggesting that people often use easily available excuses or prior rationalization as a pre-violation justification or a limit on violence. Perception of experience represents one such readily available excuse, given that political and social structures rooted in religion, culture, conservatism, and anthropocentrism [33–35] establish an order in which animals are deemed inferior, believed to possess less capacity to feel, exhibit emotions, and have less intrinsic value due to their experience capacities [67, 88–93]. Furthermore, such excuses are easily employed due to their function of prior rationalization [7]. We suggest that people use this strategy to dodge anticipated embarrassment or guilt [77–79] maintain their self-perception as moral, avoid social punishment even if they engage in instrumental use of animals [73–76] and wish to continue doing so. However, in model 1 (Fig 1), the effects were stronger than in model 2 (Fig 2), suggesting that attitudes towards animals (violence acceptance) remain relatively constant and differences in perception of experience were too small to induce changes in these attitudes, or that attitudes towards animals are the result of our intentions and behavior rather than the other way around [106, 107].

We propose that changes in violence were insignificant, despite the increase in perception of experience in the Experience 2 group versus Experience 1 (Figs 1 and 2; path $c_2$: $X_2 \rightarrow Y$), since the perception of homeostatic emotions (pain, fear, hunger, thirst—manipulated in Experience 1) is crucial in reducing violence, as they indicate suffering that humans would want animals to evade. This aligns with Bentham's [108] perspective on human-animal relationships: "The question is not, Can they *reason*? nor, Can they *talk*? but, Can they *suffer*?" (p. 122). Simultaneously, an increase in the perception of positively valenced or more complex emotions (joy, pleasure, shame, as well as personality and consciousness—manipulated in Experience 2) does not lead to an additional reduction in violence. We interpret this to mean that these emotions are linked to happiness rather than violence. Furthermore, the way mental capacities are perceived, i.e., the perception of experience capacities, is likely to be incremental. If a being is perceived to be capable of experiencing pleasure and joy more strongly and to have a personality, the perception of that being's capacity for experiences such as pain, hunger, and fear also heightens. Conversely, the opposite does not occur. The absence of a difference in violence levels, despite an increase in the perception of more complex emotions, contradicts previous studies on the infrahumanization of animals [7].

## Limitations and future directions

As for the limitations of our study, the effect of perception of experience on violence acceptance was only established as an indirect effect in model 2 (Fig 2). Another limitation pertains to the measurement of violence intentions by the Animal Matrix through a single choice. Finally, our sample consisted solely of Polish individuals. To ensure the generalizability of these findings, it is essential to replicate this study across diverse cultures and countries. For future investigations into this phenomenon, we recommend studies that: (1) allow perception of experience to be increased without changes in perception of agency; (2) lead to stronger

effects of manipulation; and (3) significance of path c (total effect). Additionally, the Animal Matrix tool should be expanded with more matrices to ensure better validity. Lastly, further research into the function of perceiving mental capacities requires the measurement of the broadest possible spectrum of these capacities to rule out alternative explanations and spurious correlations, particularly the fact that the perception of mind may serve different functions in relation to different categories of violence and animals.

## Conclusions

To sum up, our research has revealed a unique function of Perception of Experience in decreasing instrumental violence against animals. When this perception was measured concurrently with perception of agency as a mechanism of anthropomorphization, the latter was found to be ineffective. This aligns with the findings of Potocka & Bielecki [12]; which showed that only experience capacities are negatively associated with acceptance of violence against animals, as they establish moral patient status, while controlled anthropomorphization as a form of mental similarity to humans is not effective. However, this does not corroborate previous studies in the field of meat consumption that propose that anthropomorphization reduces instrumental violence [10, 11, 13, 14]. We reconcile this discrepancy in two ways: firstly, by suggesting that anthropomorphization may play a specific role in meat consumption, i.e., to avoid consuming beings that are similar to us and thus engaging in cannibalism [109–111]. Secondly, we propose that in previous studies, the function of anthropomorphization (perception of agency) was tested without controlling for perception of experience. Consequently, the perceived effect could be an illusion.

## Supporting information

**S1 Appendix. Pilot study.** Experimental manipulation.
(DOCX)

## Acknowledgments

We would like to express our sincere gratitude to our advisor Dr. Agnieszka Golec de Zavala for her invaluable guidance and support throughout the research process.

## Author Contributions

**Conceptualization:** Agnieszka Potocka.

**Data curation:** Agnieszka Potocka.

**Formal analysis:** Agnieszka Potocka.

**Funding acquisition:** Agnieszka Potocka.

**Investigation:** Agnieszka Potocka.

**Methodology:** Agnieszka Potocka, Maksymilian Bielecki.

**Project administration:** Agnieszka Potocka.

**Resources:** Agnieszka Potocka.

**Supervision:** Maksymilian Bielecki.

**Validation:** Agnieszka Potocka, Maksymilian Bielecki.

**Visualization:** Agnieszka Potocka.

**Writing – original draft:** Agnieszka Potocka, Maksymilian Bielecki.

**Writing – review & editing:** Agnieszka Potocka, Maksymilian Bielecki.

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
