## [Decision Letter · Decision Letter 0]

11 Oct 2023

PONE-D-23-19973Increased perception of the experience dimension of animal mind reduces instrumental violence against animalsPLOS ONE

Dear Dr. Potocka,

Thank you for submitting your manuscript to PLOS ONE. After careful consideration, we feel that it has merit but does not fully meet PLOS ONE’s publication criteria as it currently stands. Therefore, we invite you to submit a revised version of the manuscript that addresses the points raised during the review process.

Please find the attached reviewers comments and complete the minor revision as suggested by reviewers.

We look forward to receiving your revised manuscript.

Kind regards,

Mumtaz Alam, PhD

Academic Editor

PLOS ONE

Journal Requirements:

"Yes. This work was supported by the Animal Advocacy Research Fund, funded by Animal Charity Evaluators. Agnieszka Potocka (Principal Investigator) was the author who received the award. Agnieszka Golec de Zavala and Maksymilian Bielecki were declared as key collaborators. The grant does not have a specific grant number and was signed on September 10, 2020. Funder name: Animal Charity Evaluators https://researchfund.animalcharityevaluators.org/. Since funder may have a vested interest in the research reported in the enclosed paper to eliminate any role of this organization in the research, analysis, or paper preparation process, the study was preregistered before collecting data, and funding was not contingent upon results. " 

"I have read the journal's policy and the authors of this manuscript have the following competing interests: To conduct this study Agnieszka Potocka received Animal Advocacy Research Fund, funded by Animal Charity Evaluators."

Reviewers' comments:

Reviewer's Responses to Questions

**Comments to the Author**

1. Is the manuscript technically sound, and do the data support the conclusions?

Reviewer #1: Yes

Reviewer #2: Yes

2. Has the statistical analysis been performed appropriately and rigorously? 

Reviewer #1: Yes

Reviewer #2: Yes

3. Have the authors made all data underlying the findings in their manuscript fully available?

Reviewer #1: Yes

Reviewer #2: Yes

4. Is the manuscript presented in an intelligible fashion and written in standard English?

Reviewer #1: Yes

Reviewer #2: Yes

5. Review Comments to the Author

Reviewer #1: The author has presented the research results on the relationship between the perception of animal minds and instrumental violence against animals, transforming them into valuable global guidelines. Please refer to the attachment for suggestions and recommendations.

Reviewer #2: Comments to the Author

Line 4. Add an article to be " dimension of the animal mind".

Line 20. Please rewrite “perception of agency " to be " perceptions of agency".

Line 22. Please rewrite " task which increased perception of the experience dimension of animal" to be "task that increased their perception of the experience dimension of the animal".

Line 23. Please rewrite "check and to measure " to be "check to measure".

Line 29. Please rewrite " hunger, thirst) which " to be “hunger, and thirst), which ".

Line 34. Please rewrite " about functions " to be " about the functions ".

Line 35. Please rewrite " of the experience "to be "of experience".

Line 36. Please rewrite " rationalize violence," to be " rationalize violence".

Line 55. Please rewrite " still gap" to be " still a gap ".

Line 57. Please rewrite " nthropomorphization, or (2) they " to be " nthropomorphization, (2) they ".

Line 62. Please rewrite " against animals, and" to be " against animals and".

Line 72. Please rewrite " dairy products or eggs " to be " dairy products, or eggs “.

Line 74. Please rewrite " attitudes toward animal" to be " attitudes towards animal".

Line 75. Please add space " [29, 30]or welfare " to be " [29, 30] or welfare ".

Line 75. Please rewrite " Authors highlight " to be " The authors highlight ".

Line 76. Please rewrite " worldviews including " to be " worldviews, including ".

Line 77. Please rewrite " facilitate such instrumental " to be " facilitate the instrumental”

Line 78. Please rewrite " categories –" to be " categories:”.

Line 84. Please rewrite " [46], values " to be " [46], and values”.

Line 84. Please rewrite " obedience and security " to be " obedience, and security”.

Line 90. Please rewrite " industry as well as " to be " industry, as well as”.

Line 99. Please rewrite " constant, and" to be " constant and”.

Line 100. Please rewrite " thinking and ability.” to be " thinking, and ability”.

Line 103. Please rewrite " fish and" to be " fish, and.”.

Line 105. Remove unnecessary spaces " 65] —people fit their “to be " People fit their”.

Line 106. Please rewrite " perception of animal " to be " perception of the animal”.

Line 106. Please rewrite " intention and socially “to be " intention, and socially”.

Line 109. Please rewrite " pasture[6]. " to be " pasture [6].”.

Line 111. Please rewrite " entities which have less capacities to feel “to be " entities that have less capacity to feel”.

Line 113. Please rewrite " perception of animal " to be " perception of the animal”.

Line 116. Change the preposition “unique for humans” to be “unique to humans”.

Line 117. Please rewrite " nonhumans”[p. 221, 22], “to be " nonhumans [p. 221, 22],”.

Line 118. Please rewrite " phenomena and" to be " phenomena, and”.

Line 119. Please rewrite " defined in similar fashion—mainly in law, philosophy or in literature “to be " defined in a similar fashion—mainly in law, philosophy, or literature”.

Lines 122, 123. Please rewrite " important is anthropomorphization’s role in limiting violence, it is certainly not the only aspect of the animal mind perception regulating our moral choices " to be " important anthropomorphization’s role in limiting violence is, it is certainly not the only aspect of animal mind perception regulating our moral choices”.

Line 136. Please rewrite " Gray et al. [17] which “to be " Gray et al. [17], which”.

Line 138. Please rewrite " observer—as is " to be " observer, as”.

Line 141. Please rewrite " compensation, " to be " compensation”.

Line 144. Please rewrite " (moral patient)" to be " (a moral patient)”.

Line 148. Please rewrite " moral agents and" to be " moral agents, and”.

Lines 151, 152. Please rewrite " victim to make a shift from moral act (the victim is a moral patient) to not moral act (the victim is not a moral patient but an inanimate product, object or thing)" to be " victim by making a shift from a moral act (the victim is a moral patient) to a non-moral act (the victim is not a moral patient but an inanimate product, object, or thing).”.

Line 153. Please rewrite " dismissing of moral" to be " dismissal of moral”.

Line 158. Please rewrite " since from the perspective of moral agent " to be " since, from the perspective of moral agents”.

Line 164. Please rewrite " religion[35, 87, 88] and" to be " religion [35, 87, 88], and”.

Line 172. Please rewrite " dimension," to be " dimension”.

Line 177. Please rewrite " [17], that" to be " [17] that”.

Line 178. Please rewrite " thus, " to be " thus”.

Line 181. Please rewrite " [12], " to be " [12]”.

Line 185. Please rewrite " animals, " to be " animals”.

Line 188. Please rewrite " of animal mind, and" to be " of the animal mind and”.

Line 191. Please rewrite " of animal mind " to be " of the animal mind”.

Line 192. Please rewrite " animals and " to be " animals, and”.

Lines 196, 197. Please rewrite " and expected that manipulation would increase perception of experience capacities in Experience1" to be " and it was expected that manipulation would increase perceptions of experience capacities in the Experience1”.

Line 207. Please rewrite " 2/2022 – “to be " 2/2022;”.

Line 212. Please rewrite " adult people " to be " adults”.

Line 214. Please rewrite " survey, description " to be " survey, a description”.

Line 216. Please rewrite " was displayed." to be " were displayed.”.

Line 217. Please rewrite " on the question" to be " to the question”.

Line 219. Please rewrite " electronically and" to be " electronically, and”.

Line 221. Please rewrite " Warsaw approved" to be " Warsaw, approved”.

Line 222. Please rewrite " Experience1, Experience2, and" to be " Experience 1, Experience 2, and”.

Line 223. Please rewrite " they performed " to be " performed”.

Line 225. Please rewrite " Authors" to be " The authors”.

Line 235. Remove the redundant word “in in the country,” to be " in the country,”.

Line 236. Please rewrite " education, other" to be " education, and other”.

Line 237. Please rewrite " years and " to be " years, and”.

Line 241-243. Please rewrite " which were assigned to their specific experimental condition. In the Control group, the text was about colors. In Experience1, it was about cows’ and pigs’ capacities to feel pain, fear, hunger." to be "that were assigned to their specific experimental condition. In the control group, the text was about colors. In Experience 1, it was about cows’ and pigs’ capacities to feel pain, fear, and hunger.”.

Line 244-248. Please rewrite " joy, pleasure, and about the animals' personality. Next, the participants were asked to write a short post for Facebook or Twitter about the text they read and to complete the Animal Mind Questionnaire (AMQ-PL, [12], based on the Mind Survey [15]. We used the AMQ-PL subscale of Experience (a = 0.90) to verify the effectiveness of the manipulation and the Agency subscale (a = 0.89) to control for Anthropomorphization. " to be " joy, and pleasure, and about the animals' personalities. Next, the participants were asked to write a short post for Facebook or Twitter about the text they read and to complete the Animal Mind Questionnaire (AMQ-PL) based on the Mind Survey [15]. We used the AMQ-PL subscale of experience (a = 0.90) to verify the effectiveness of the manipulation and the agency subscale (a = 0.89) to control for anthropomorphization.”.

Line 250-256. Please rewrite " The Agency subscale uses items on morality, memory, emotion recognition, planning, self-control, communication, and thought. Participants instructions were following: “Using a scale of 0 (not at all) to 6 (very much), please tell us how capable do you think horses, cows, goats, sheep are of:” and provided with descriptions of 18 mental capacities from the original scale, e.g., “understanding how others are feeling”, “feeling afraid or fearful”, “telling right from wrong and trying to do the right thing”. " to be " The agency subscale uses items on morality, memory, emotion recognition, planning, self-control, communication, and thought. Participants instructions were the following: “Using a scale of 0 (not at all) to 6 (very much), please tell us how capable you think horses, cows, goats, and sheep are of:” and provided with descriptions of 18 mental capacities from the original scale, e.g., “understanding how others are feeling”, “feeling afraid or fearful" and “telling right from wrong and trying to do the right thing”.”.

Line 259. Please rewrite " We measured intentions" to be " We measured the intentions”.

Line 261. Please rewrite " [98] and" to be " [98], and”.

Line 265. Please rewrite " maintenance and providing food, facilities, veterinary care and " to be " maintaining and providing food, facilities, veterinary care, and”.

Line 271. Please rewrite " item " to be " items”.

Line 273. Please rewrite " cost–profit ratio" to be " cost-profit ratio”.

Line 283. Please rewrite " consumption”," to be " consumption" and”.

Line 289. Please rewrite " by according" to be " according to the”.

Line 290. Please rewrite " 102]," to be " 102]”.

Lines 291, 292. Please rewrite " a = 0.75, in order to increase Fairness subscale reliability, item 27 was excluded, " to be " a = 0.75; in order to increase fairness subscale reliability, item 27 was excluded”.

Line 295. Please rewrite " we calculated means " to be " we calculated the means”.

Line 237. Please rewrite " in OLS model for Perception of Experience, Anthropomorphization" to be " in the OLS model for perception of experience, anthropomorphization,”.

Lines 300-304. Please rewrite " Perception of Experience in the experimental groups (Experience1 + Experience2) versus Control, however, this is accompanied by higher Anthropomorphization (Contrast1). A contrast test between the experimental groups (Contrast2) revealed higher Perception of Experience in Experience2 than in Experience1, without differences in Anthropomorphization. " to be " perceptions of experience in the experimental groups (Experience 1 + Experience 2) versus control; however, this is accompanied by higher anthropomorphization (Contrast 1). A contrast test between the experimental groups (Contrast 2) revealed a higher perception of experience in Experience 2 than in Experience 1, without differences in anthropomorphization.”.

Line 309. Please rewrite " Contrast1 = both experimental groups together (Experience1 and Experience 2) vs the Control group, contrast2 = Experience2 vs Experience1." to be " Contrast 1 = both experimental groups together (Experience 1 and Experience 2) vs. the control group; contrast 2 = Experience 2 vs. Experience 1.”.

Line 311. Please rewrite " (Fig 1 and 2) which" to be " (Figs. 1 and 2), which”.

Line 315. Please rewrite " model 1 and " to be " model 1, and”.

Line 317. Please rewrite " (Experience1 + Experience2) with the Control group (X1) and the experimental groups with each other (Experience2 vs Experience1, X2)." to be " (Experience 1 + Experience 2) with the control group (X1) and the experimental groups with each other (Experience 2 vs. Experience 1, X2).”.

Line 290. Please rewrite " 102]," to be " 102]”.

Lines 291, 292. Please rewrite " a = 0.75, in order to increase Fairness subscale reliability, item 27 was excluded, " to be " a = 0.75; in order to increase fairness subscale reliability, item 27 was excluded”.

Line 295. Please rewrite " we calculated means " to be " we calculated the means”.

Line 331. Please rewrite " (Figs 1 and 2)" to be " (Figs. 1 and 2)”.

Line 332. Please rewrite " animals and " to be " animals, and”.

Line 334. Please rewrite " upon revealing the total and indirect effects for the outcome variable Violence Intentions (Fig 2)" to be " by revealing the total and indirect effects of the outcome variable Violence Intentions (Fig. 2)”.

Line 338. Please rewrite " Y , Fig 2), " to be " Y, Fig. 2)”.

Line 341. Please rewrite " Y - Figs 1 and 2) " to be " Y, Figs. 1 and 2”.

Line 346. Please rewrite " Moral Patient," to be " moral patients,”.

Line 347. Please rewrite " have less moral" to be " have fewer moral”.

Lines 356-358. Please rewrite " [77 – 79] maintain their self-perception as moral, avoid social punishment even if they engage in instrumental use of animals [73 - 76] " to be " [77–79] maintain their self-perception as moral, avoid social punishment even if they engage in the instrumental use of animals [73-76]”.

Line 360. Please rewrite " constant, and differences in Perception of Experience were " to be " constant and differences in Perception of Experience are”.

Lines 364-366. Please rewrite " Experience2 group versus Experience1 (Fig 1 and 2 path c2: X2 → Y) since the perception of homeostatic emotions (pain, fear, hunger, thirst - manipulated in Experience1) " to be " Experience 2 group versus Experience 1 (Figs. 1 and 2; path c2: X2 → Y), since the perception of homeostatic emotions (pain, fear, hunger, and thirst—manipulated in Experience 1)”.

nor, Can they talk?

Line 368. Please rewrite " nor, Can they talk? " to be " Nor, can they talk?)”.

Line 374. Please rewrite " strongly," to be " strongly”.

Line 376. Please rewrite " absence of difference” to be " absence of a difference”.

Line 381. Please rewrite " (Fig 2). " to be " (Fig. 2).”.

Line 386-387. Please rewrite " Agency, (2) lead to stronger effects of manipulation " to be " agency; (2) lead to stronger effects of manipulation;”.

Line 392. Please rewrite " animals" to be " animals.”.

Line 396. Please rewrite " Perception of Agency” to be " the perception of agency”.

Line 397. Please rewrite " [12] " to be " [12],”.

Line 401. Please rewrite " propose anthropomorphization" to be " propose that anthropomorphization”.

Line 404. Please rewrite " [109 -111].” to be " [109-111].”.

6. PLOS authors have the option to publish the peer review history of their article (what does this mean?). If published, this will include your full peer review and any attached files.

Reviewer #1: **Yes: **Yudi Adinata

Reviewer #2: No

---

## [Author Response · Author response to Decision Letter 0]

27 Oct 2023

Responses to Editor:

Response:

This has been checked. The supporting information has been changed in manuscript.

2. Thank you for stating in your Funding Statement: (…)

Please provide an amended statement that declares *all* the funding or sources of support (whether external or internal to your organization) received during this study, as detailed online in our guide for authors at http://journals.plos.org/plosone/s/submit-now. (…) Please include your amended Funding Statement within your cover letter. We will change the online submission form on your behalf. 

Response: 

Yes. This work was supported by the Animal Advocacy Research Fund, funded by Animal Charity Evaluators. Agnieszka Potocka (Principal Investigator) was the author who received the award. Agnieszka Golec de Zavala and Maksymilian Bielecki were declared as key collaborators. The grant does not have a specific grant number and was signed on September 10, 2020. Funder name: Animal Charity Evaluators https://researchfund.animalcharityevaluators.org/. Since funder may have a vested interest in the research reported in the enclosed paper to eliminate any role of this organization in the research, analysis, or paper preparation process, the study was preregistered before collecting data, and funding was not contingent upon results. All funds or sources of support received during this study have been described in accordance with the requirements in the online author's guide at http://journals.plos.org/plosone/s/submit-now. The funders had no role in study design, data collection and analysis, decision to publish, or preparation of the manuscript. There was no additional external funding received for this study.

"I have read the journal's policy and the authors of this manuscript have the following competing interests: To conduct this study Agnieszka Potocka received Animal Advocacy Research Fund, funded by Animal Charity Evaluators."

Please confirm that this does not alter your adherence to all PLOS ONE policies on sharing data and materials.

Response:

I have read the journal's policy and the authors of this manuscript have the following competing interests: To conduct this study Agnieszka Potocka received Animal Advocacy Research Fund, funded by Animal Charity Evaluators.

This does not alter our adherence to PLOS ONE policies on sharing data and materials. The data and protocol are licensed under CC-By Attribution 4.0 International.

Response:

The DOI for prerefistration and database has been added in paper:

“The study was preregistered on OSF: https://doi.org/10.17605/OSF.IO/6JKST (database: https://doi.org/10.17605/OSF.IO/XMYV8)”

Data Availability statement:

All data are fully available on OSF: study preregistration: https://doi.org/10.17605/OSF.IO/6JKST, database, and syntax: https://doi.org/10.17605/OSF.IO/XMYV8, licensed under CC-By Attribution 4.0 International.

5. Please review your reference list to ensure that it is complete and correct.

Response:

All references has been checked to be complete and correct.

Responses to Reviewers’.

The manuscript has been checked and the necessary changes have been made in accordance with the reviewers’ suggestions. The responses to all comments have been prepared and given below.

Reviewer #1

1. Abstract: The abstract should offer a concise summary of your study, including the research question, methodology, key findings, and implications. Ensure that it highlights the significance of your research. Make sure your abstract provides a clear overview of the study.

Response:

The abstract provides a clear overview of the study including the research question, methodology, key findings, and implications.

2. Introduction: Clearly state the research problem and research questions in the introduction. Provide a brief literature review to establish the context and importance of your study. Simplify the introduction for better readability. Provide a brief literature review to establish the context and importance of your study.

Response: 

The introduction meets these requirements.

3. Methods: Explain your experimental design in detail, including participant recruitment, the procedure they followed, and the manipulation task. Provide a clear rationale for your choice of measures and questionnaires.

Response: 

The methods meets these requirements.

4. Results: Present your results in a clear and organized manner. Avoid overwhelming the reader with excessive statistical details. Focus on the most critical results that address your hypotheses.

Response: 

The results meets these requirements.

5. Discussion: Discuss the effect sizes and practical significance of your findings. Explore the implications of your findings for the field and for addressing the issue of instrumental violence against animals. Address potential limitations of your study, such as sample size, generalizability, or other methodological concerns. Suggest areas for future research or extensions of your work.

Response: 

The discussion meets these requirements.

6. Conclusions: Conclude your paper by summarizing the main findings and their significance. Discuss the practical implications of your research and how it contributes to the broader understanding of the topic. Please provide a description of the scientific novelty presented in the study, including a section on the study’s limitations and research perspectives in the conclusion.

Response: 

The conclusion meets these requirements.

7. Lastly, it’s advisable to include the most recent references, primarily those published between 2017 and 2023, except in the method section where older references might be more relevant.

Response

The relationship between humans and animals is not a central area of study in social psychology, which is why it may not be comprehensively covered in the most recent publications:

50% references are published 2007-2023

11% references are published 2019-2023

Reviewer #2

1. We have considered the following comments, and as a result, the text has been revised in accordance with the reviewer's suggestion:

Line 4. Add an article to be " dimension of the animal mind".

Line 20. Please rewrite “perception of agency " to be " perceptions of agency".

Line 22. Please rewrite " task which increased perception of the experience dimension of animal" to be "task that increased their perception of the experience dimension of the animal".

Line 23. Please rewrite "check and to measure " to be "check to measure".

Line 29. Please rewrite " hunger, thirst) which " to be “hunger, and thirst), which ".

Line 34. Please rewrite " about functions " to be " about the functions ".

Line 35. Please rewrite " of the experience "to be "of experience".

Line 36. Please rewrite " rationalize violence," to be " rationalize violence".

Line 55. Please rewrite " still gap" to be " still a gap ".

Line 57. Please rewrite " nthropomorphization, or (2) they " to be " nthropomorphization, (2) they ".

Line 62. Please rewrite " against animals, and" to be " against animals and".

Line 72. Please rewrite " dairy products or eggs " to be " dairy products, or eggs “.

Line 74. Please rewrite " attitudes toward animal" to be " attitudes towards animal".

Line 75. Please add space " [29, 30]or welfare " to be " [29, 30] or welfare ".

Line 75. Please rewrite " Authors highlight " to be " The authors highlight ".

Line 76. Please rewrite " worldviews including " to be " worldviews, including ".

Line 77. Please rewrite " facilitate such instrumental " to be " facilitate the instrumental”

Line 78. Please rewrite " categories –" to be " categories:”.

Line 84. Please rewrite " [46], values " to be " [46], and values”.

Line 84. Please rewrite " obedience and security " to be " obedience, and security”.

Line 90. Please rewrite " industry as well as " to be " industry, as well as”.

Line 99. Please rewrite " constant, and" to be " constant and”.

Line 100. Please rewrite " thinking and ability.” to be " thinking, and ability”.

Line 103. Please rewrite " fish and" to be " fish, and.”.

Line 105. Remove unnecessary spaces " 65] —people fit their “to be " People fit their”.

Line 106. Please rewrite " perception of animal " to be " perception of the animal”.

Line 106. Please rewrite " intention and socially “to be " intention, and socially”.

Line 109. Please rewrite " pasture[6]. " to be " pasture [6].”.

Line 111. Please rewrite " entities which have less capacities to feel “to be " entities that have less capacity to feel”.

Line 113. Please rewrite " perception of animal " to be " perception of the animal”.

Line 116. Change the preposition “unique for humans” to be “unique to humans”.

Line 117. Please rewrite " nonhumans”[p. 221, 22], “to be " nonhumans [p. 221, 22],”.

Line 118. Please rewrite " phenomena and" to be " phenomena, and”.

Line 119. Please rewrite " defined in similar fashion—mainly in law, philosophy or in literature “to be " defined in a similar fashion—mainly in law, philosophy, or literature”.

Lines 122, 123. Please rewrite " important is anthropomorphization’s role in limiting violence, it is certainly not the only aspect of the animal mind perception regulating our moral choices " to be " important anthropomorphization’s role in limiting violence is, it is certainly not the only aspect of animal mind perception regulating our moral choices”.

Line 136. Please rewrite " Gray et al. [17] which “to be " Gray et al. [17], which”.

Line 138. Please rewrite " observer—as is " to be " observer, as”.

Line 141. Please rewrite " compensation, " to be " compensation”.

Line 144. Please rewrite " (moral patient)" to be " (a moral patient)”.

Line 148. Please rewrite " moral agents and" to be " moral agents, and”.

Lines 151, 152. Please rewrite " victim to make a shift from moral act (the victim is a moral patient) to not moral act (the victim is not a moral patient but an inanimate product, object or thing)" to be " victim by making a shift from a moral act (the victim is a moral patient) to a non-moral act (the victim is not a moral patient but an inanimate product, object, or thing).”.

Line 153. Please rewrite " dismissing of moral" to be " dismissal of moral”.

Line 158. Please rewrite " since from the perspective of moral agent " to be " since, from the perspective of moral agents”.

Line 164. Please rewrite " religion[35, 87, 88] and" to be " religion [35, 87, 88], and”.

Line 172. Please rewrite " dimension," to be " dimension”.

Line 177. Please rewrite " [17], that" to be " [17] that”.

Line 178. Please rewrite " thus, " to be " thus”.

Line 181. Please rewrite " [12], " to be " [12]”.

Line 185. Please rewrite " animals, " to be " animals”.

Line 188. Please rewrite " of animal mind, and" to be " of the animal mind and”.

Line 191. Please rewrite " of animal mind " to be " of the animal mind”.

Line 192. Please rewrite " animals and " to be " animals, and”.

Lines 196, 197. Please rewrite " and expected that manipulation would increase perception of experience capacities in Experience1" to be " and it was expected that manipulation would increase perceptions of experience capacities in the Experience1”.

Line 207. Please rewrite " 2/2022 – “to be " 2/2022;”.

Line 212. Please rewrite " adult people " to be " adults”.

Line 214. Please rewrite " survey, description " to be " survey, a description”.

Line 216. Please rewrite " was displayed." to be " were displayed.”.

Line 217. Please rewrite " on the question" to be " to the question”.

Line 219. Please rewrite " electronically and" to be " electronically, and”.

Line 221. Please rewrite " Warsaw approved" to be " Warsaw, approved”.

Line 222. Please rewrite " Experience1, Experience2, and" to be " Experience 1, Experience 2, and”.

Line 223. Please rewrite " they performed " to be " performed”.

Line 225. Please rewrite " Authors" to be " The authors”.

Line 235. Remove the redundant word “in in the country,” to be " in the country,”.

Line 236. Please rewrite " education, other" to be " education, and other”.

Line 237. Please rewrite " years and " to be " years, and”.

Line 241-243. Please rewrite " which were assigned to their specific experimental condition. In the Control group, the text was about colors. In Experience1, it was about cows’ and pigs’ capacities to feel pain, fear, hunger." to be "that were assigned to their specific experimental condition. In the control group, the text was about colors. In Experience 1, it was about cows’ and pigs’ capacities to feel pain, fear, and hunger.”.

Line 244-248. Please rewrite " joy, pleasure, and about the animals' personality. Next, the participants were asked to write a short post for Facebook or Twitter about the text they read and to complete the Animal Mind Questionnaire (AMQ-PL, [12], based on the Mind Survey [15]. We used the AMQ-PL subscale of Experience (a = 0.90) to verify the effectiveness of the manipulation and the Agency subscale (a = 0.89) to control for Anthropomorphization. " to be " joy, and pleasure, and about the animals' personalities. Next, the participants were asked to write a short post for Facebook or Twitter about the text they read and to complete the Animal Mind Questionnaire (AMQ-PL) based on the Mind Survey [15]. We used the AMQ-PL subscale of experience (a = 0.90) to verify the effectiveness of the manipulation and the agency subscale (a = 0.89) to control for anthropomorphization.”.

we left [12] 

Line 250-256. Please rewrite " The Agency subscale uses items on morality, memory, emotion recognition, planning, self-control, communication, and thought. Participants instructions were following: “Using a scale of 0 (not at all) to 6 (very much), please tell us how capable do you think horses, cows, goats, sheep are of:” and provided with descriptions of 18 mental capacities from the original scale, e.g., “understanding how others are feeling”, “feeling afraid or fearful”, “telling right from wrong and trying to do the right thing”. " to be " The agency subscale uses items on morality, memory, emotion recognition, planning, self-control, communication, and thought. Participants instructions were the following: “Using a scale of 0 (not at all) to 6 (very much), please tell us how capable you think horses, cows, goats, and sheep are of:” and provided with descriptions of 18 mental capacities from the original scale, e.g., “understanding how others are feeling”, “feeling afraid or fearful" and “telling right from wrong and trying to do the right thing”.”.

Line 259. Please rewrite " We measured intentions" to be " We measured the intentions”.

Line 261. Please rewrite " [98] and" to be " [98], and”.

Line 265. Please rewrite " maintenance and providing food, facilities, veterinary care and " to be " maintaining and providing food, facilities, veterinary care, and”.

Line 271. Please rewrite " item " to be " items”.

Line 273. Please rewrite " cost–profit ratio" to be " cost-profit ratio”.

Line 283. Please rewrite " consumption”," to be " consumption" and”.

Line 289. Please rewrite " by according" to be " according to the”.

Line 290. Please rewrite " 102]," to be " 102]”.

Lines 291, 292. Please rewrite " a = 0.75, in order to increase Fairness subscale reliability, item 27 was excluded, " to be " a = 0.75; in order to increase fairness subscale reliability, item 27 was excluded”.

Line 295. Please rewrite " we calculated means " to be " we calculated the means”.

Line 237. Please rewrite " in OLS model for Perception of Experience, Anthropomorphization" to be " in the OLS model for perception of experience, anthropomorphization,”.

Lines 300-304. Please rewrite " Perception of Experience in the experimental groups (Experience1 + Experience2) versus Control, however, this is accompanied by higher Anthropomorphization (Contrast1). A contrast test between the experimental groups (Contrast2) revealed higher Perception of Experience in Experience2 than in Experience1, without differences in Anthropomorphization. " to be " perceptions of experience in the experimental groups (Experience 1 + Experience 2) versus control; however, this is accompanied by higher anthropomorphization (Contrast 1). A contrast test between the experimental groups (Contrast 2) revealed a higher perception of experience in Experience 2 than in Experience 1, without differences in anthropomorphization.”.

Line 309. Please rewrite " Contrast1 = both experimental groups together (Experience1 and Experience 2) vs the Control group, contrast2 = Experience2 vs Experience1." to be " Contrast 1 = both experimental groups together (Experience 1 and Experience 2) vs. the control group; contrast 2 = Experience 2 vs. Experience 1.”.

Line 315. Please rewrite " model 1 and " to be " model 1, and”.

Line 317. Please rewrite " (Experience1 + Experience2) with the Control group (X1) and the experimental groups with each other (Experience2 vs Experience1, X2)." to be " (Experience 1 + Experience 2) with the control group (X1) and the experimental groups with each other (Experience 2 vs. Experience 1, X2).”.

Line 290. Please rewrite " 102]," to be " 102]”.

Lines 291, 292. Please rewrite " a = 0.75, in order to increase Fairness subscale reliability, item 27 was excluded, " to be " a = 0.75; in order to increase fairness subscale reliability, item 27 was excluded”.

Line 295. Please rewrite " we calculated means " to be " we calculated the means”.

Line 332. Please rewrite " animals and " to be " animals, and”.

Line 346. Please rewrite " Moral Patient," to be " moral patients,”.

Line 347. Please rewrite " have less moral" to be " have fewer moral”.

Lines 356-358. Please rewrite " [77 – 79] maintain their self-perception as moral, avoid social punishment even if they engage in instrumental use of animals [73 - 76] " to be " [77–79] maintain their self-perception as moral, avoid social punishment even if they engage in the instrumental use of animals [73-76]”.

Line 360. Please rewrite " constant, and differences in Perception of Experience were " to be " constant and differences in Perception of Experience are”.

Lines 364-366. Please rewrite " Experience2 group versus Experience1 (Fig 1 and 2 path c2: X2 → Y) since the perception of homeostatic emotions (pain, fear, hunger, thirst - manipulated in Experience1) " to be " Experience 2 group versus Experience 1 (Figs. 1 and 2; path c2: X2 → Y), since the perception of homeostatic emotions (pain, fear, hunger, and thirst—manipulated in Experience 1)”.

Line 374. Please rewrite " strongly," to be " strongly”.

Line 376. Please rewrite " absence of difference” to be " absence of a difference”.

Line 386-387. Please rewrite " Agency, (2) lead to stronger effects of manipulation " to be " agency; (2) lead to stronger effects of manipulation;”.

Line 392. Please rewrite " animals" to be " animals.”.

Line 396. Please rewrite " Perception of Agency” to be " the perception of agency”.

Line 397. Please rewrite " [12] " to be " [12],”.

Line 401. Please rewrite " propose anthropomorphization" to be " propose that anthropomorphization”.

Line 404. Please rewrite " [109 -111].” to be " [109-111].”.

2. We have considered the following comments, and as a result, the text has been partially revised.

 • Line 311. Please rewrite " (Fig 1 and 2) which" to be " (Figs. 1 and 2), which”.

Response:

According to the author guidelines: https://journals.plos.org/plosone/s/submission-guidelines and PLOS ONE style templates: https://journals.plos.org/plosone/s/file?id=wjVg/PLOSOne_formatting_sample_main_body.pdf it has been changed to:

“(Figs 1 and 2), which”

 • Line 334. Please rewrite " upon revealing the total and indirect effects for the outcome variable Violence Intentions (Fig 2)" to be " by revealing the total and indirect effects of the outcome variable Violence Intentions (Fig. 2)”.

Response: 

As per the author guidelines found at https://journals.plos.org/plosone/s/submission-guidelines and the PLOS ONE style templates provided in https://journals.plos.org/plosone/s/file?id=wjVg/PLOSOne_formatting_sample_main_body.pdf, figures should continue to be cited as "Fig 1," "Fig 2,", so it has been changed to:

“by revealing the total and indirect effects of the outcome variable Violence Intentions (Fig 2)”

 • Line 338. Please rewrite " Y , Fig 2), " to be " Y, Fig. 2)”.

Response:

As per the author guidelines found at https://journals.plos.org/plosone/s/submission-guidelines and the PLOS ONE style templates provided in https://journals.plos.org/plosone/s/file?id=wjVg/PLOSOne_formatting_sample_main_body.pdf, figures should continue to be cited as "Fig 1," "Fig 2,", it has been changed to:

" Y, Fig 2”

 • Line 341. Please rewrite Y - Figs 1 and 2) " to be " Y, Figs. 1 and 2”

Response: 

As per the author guidelines found at https://journals.plos.org/plosone/s/submission-guidelines and the PLOS ONE style templates provided in https://journals.plos.org/plosone/s/file?id=wjVg/PLOSOne_formatting_sample_main_body.pdf, figures should continue to be cited as "Fig 1," "Fig 2,", so it has been changed to: “Y, Figs 1 and 2”. 

3. We have considered the following comments, and as a result, the text has not been revised.

 • Line 331. Please rewrite " (Figs 1 and 2)" to be " (Figs. 1 and 2).

Response: 

As per the author guidelines found at https://journals.plos.org/plosone/s/submission-guidelines and the PLOS ONE style templates provided in https://journals.plos.org/plosone/s/file?id=wjVg/PLOSOne_formatting_sample_main_body.pdf, figures should continue to be cited as "Fig 1," "Fig 2,", so it has not been changed. 

 • Line 368. Please rewrite " nor, Can they talk? " to be " Nor, can they talk?)”.

Response:

The original spelling has been preserved as it is a quoted passage.

 • Line 381. Please rewrite " (Fig 2). " to be " (Fig. 2).”.

Response: 

As per the author guidelines found at https://journals.plos.org/plosone/s/submission-guidelines and the PLOS ONE style templates provided in https://journals.plos.org/plosone/s/file?id=wjVg/PLOSOne_formatting_sample_main_body.pdf, figures should continue to be cited as "Fig 1," "Fig 2,", so it has not been changed. 

4. In response to the reviewer's feedback, which recommended converting, name of control group, variables’ and subscales’ names to lowercase letters, we implemented a standard format, ensuring all variables and subscales were consistently represented in lowercase letters.

 • Line 222

“Control, Experience 1” to be “control, Experience 1”

 • Line 249

“Experience subscale” to be “experience subscale” 

 • Lines: 259, 315, 322, 336, 385

“Violence Intentions to be “violence intentions”

 • Line 259, 281, 285, 287, 316, 328, 361, 383-384

“Violence Acceptance” to be “violence acceptance”

 • Line 291-292

“Care � = 0.72, Authority � = 0.70, Loyalty � = 0.68, Purity � = 0.73, Fairness 5-item scale � = 0.75, in order to increase fairness”

to be

“care � = 0.72, authority � = 0.70, loyalty � = 0.68, purity � = 0.73, fairness 5-item scale � = 0.75, in order to increase fairness”

 • Line 190, 304, 314, 322, 328, 333, 344, 348-349, 362, 365, 383, 389, 397, 409

“Perception of Experience” to be “perception of experience”

 • Line 307

“moral recipient” to be “moral patient”

 • Table 1, Figures 1 and 2

“Perception of Experience (Moral Patient)” to be “Perception of experience (moral patient)”

“Perception of Agency (Anthropomorphization)” to be “Perception of agency (anthropomorphization)”

 • Line 314, 334, Figures 1 and 2

“Moral patient” to be “moral patient”

 • Line 314-315, 335, 389, 409

“Perception of Agency” to be “perception of agency”

 • Line 192, 315, 322, 328, 335

“Anthropomorphization” to be “anthropomorphization”

 • Line 352, Figures 1 and 2

Perception of Experience” to be “Perception of experience”

 • Figures 1 and 2

“Violence Intention” to be “Violence intention”

“Anthropomorphization” to be “anthropomorphization”

 • Figures 1 and 2

“Experience 1 + Experience 2 vs Control” to be “ Experience 1 + Experience 2 vs, control”

 • We have implemented a standard format in Appendix.

5. On line 202, we have corrected 'path b M → X' to 'path b M→Y' as it was originally a mistake.

---

## [Decision Letter · Decision Letter 1]

15 Nov 2023

Increased perception of the experience dimension of the animal mind reduces instrumental violence against animals

PONE-D-23-19973R1

Dear Dr. Agnieszka Potocka,

We’re pleased to inform you that your manuscript has been judged scientifically suitable for publication and will be formally accepted for publication once it meets all outstanding technical requirements.

Kind regards,

Mumtaz Alam, PhD

Academic Editor

PLOS ONE

Additional Editor Comments (optional):

Accepted 

Reviewers' comments:

Reviewer's Responses to Questions

**Comments to the Author**

1. If the authors have adequately addressed your comments raised in a previous round of review and you feel that this manuscript is now acceptable for publication, you may indicate that here to bypass the “Comments to the Author” section, enter your conflict of interest statement in the “Confidential to Editor” section, and submit your "Accept" recommendation.

Reviewer #1: All comments have been addressed

Reviewer #2: All comments have been addressed

2. Is the manuscript technically sound, and do the data support the conclusions?

Reviewer #1: Yes

Reviewer #2: Yes

3. Has the statistical analysis been performed appropriately and rigorously? 

Reviewer #1: Yes

Reviewer #2: Yes

4. Have the authors made all data underlying the findings in their manuscript fully available?

Reviewer #1: Yes

Reviewer #2: Yes

5. Is the manuscript presented in an intelligible fashion and written in standard English?

Reviewer #1: Yes

Reviewer #2: Yes

6. Review Comments to the Author

Reviewer #1: The authors have implemented substantial enhancements to the manuscript, effectively addressing the suggestions provided. I am pleased to acknowledge their dedicated efforts in significantly improving the quality of this paper. Their diligent work and commitment to excellence are highly appreciated.

Reviewer #2: (No Response)

7. PLOS authors have the option to publish the peer review history of their article (what does this mean?). If published, this will include your full peer review and any attached files.

Reviewer #1: **Yes: **Yudi Adinata

Reviewer #2: No

---

## [Editor Report · Acceptance letter]

21 Nov 2023

PONE-D-23-19973R1 

Increased perception of the experience dimension of the animal mind reduces instrumental violence against animals 

Dear Dr. Potocka:

I'm pleased to inform you that your manuscript has been deemed suitable for publication in PLOS ONE. Congratulations! Your manuscript is now with our production department. 

Kind regards, 

on behalf of

Dr. Mumtaz Alam 

Academic Editor

PLOS ONE